

# Elucidation of the adaptability of *Cinnamomum camphora* seedlings from different provenances *via* analyzing photosynthetic characteristics and anatomical structure

Yufeng Liu[1,2,*], Ruobing Zou[1,2,*], Yanmeng Guo[1,2], Yu Ji[1,2], Mengyuan Li[1,2] and Yang You[1,2]

[1] Henan Institute of Science and Technology, Xinxiang, China

[2] Henan Institute of Science and Technology, Henan Engineering Technology Research Center of Characteristic Horticultural Plants Development and Utilization, Xinxiang, China

[*] These authors contributed equally to this work.

Corresponding author
Yang You, Youyang1028@126.com

## ABSTRACT

Understanding intraspecific variation in photosynthetic capacity and leaf structure is critical for optimizing provenance selection of *Cinnamomum camphora* in afforestation and urban landscaping programs. In this study, we assessed the adaptability of *C. camphora* seedlings from four provenances (Fuzhou, Wuhan, Shenzhen, and Shanghai) by examining their physiological and anatomical traits under a common garden environment in Henan Province. A total of $n = 80$ three-year-old seedlings (twenty per provenance) were evaluated for photosynthetic parameters, stomatal characteristics, and leaf structural features. The results revealed that the net photosynthetic rates of *C. camphora* seedlings from four different provenance sites exhibited a "double-peak" curve and a photosynthetic "lunch break" phenomenon, which was strongly positively correlated with stomatal conductance, transpiration rate, and Soil Plant Analysis Development (SPAD) value. Significant differences in stomatal characteristics were observed among the seedlings from four provenances. The seedlings from Fuzhou and Wuhan exhibited larger stomatal width, area, resulting in superior stomatal gas exchange than that in the seedlings from other provenances. Conversely, *C. camphora* seedlings from Shanghai exhibited smaller stomatal area and density, indicating poorer gas exchange and reduced adaptability. The stem cortex cells, stem phloem, stem pith diameter, leaf palisade tissue thickness, and leaf thickness of the seedlings from Shanghai were significantly lower than those of the seedlings from other provenances, indicating that these structural characteristics do not exhibit any photosynthetic advantages over other provenances. In contrast, the seedlings from Fuzhou and Wuhan exhibited larger stem pith diameter, thicker mesophyll cell, and greater leaf thickness, which enhanced their photosynthetic capabilities. Among the seedlings from the four different origins, those from Fuzhou and Wuhan exhibited the best overall photosynthetic ability and strongest adaptability. Conversely, the seedlings from Shanghai exhibited the poorest overall photosynthetic ability and weakest adaptability. Despite similarities in climate, the environmental conditions of different provenances did not appear to have a

significant correlation with leaf anatomy. This study provided valuable insights for the introduction of *C. camphora* in various regions in China.

## INTRODUCTION

Plant species with widespread distribution typically exhibit a high degree of intraspecific variation in functional traits, which is reflected in their phenotypic diversity. This diversity may originate from genetic differences arising because of local adaptation and phenotypic plasticity (*Ren et al., 2020*). Investigating the environmental adaptability of plants has a significant practical value for their conservation, cultivation, and introduction. To disentangle the effects of genetic and environmental factors on plant physiological and ecological traits, researchers commonly employ common garden experiments, where individuals from different provenances are grown under uniform environmental conditions. This approach minimizes environmental noise and allows for a robust assessment of inherited variation and trait-based adaptability (*Albaugh et al., 2018*; *Nicotra et al., 2010*; *Valladares, Gianoli & Gómez, 2007*). Furthermore, it provides valuable insights into physiological processes such as gas exchange, water use efficiency, and stress tolerance, which are critical for understanding plant responses to climate variation.

For many years, the study of plant photosynthesis has been a major focus in ecology, plant physiology, and environmental restoration (*Yang, Yang & Gao, 2023*). Photosynthesis is a critical process through which plants metabolize, accumulate metabolites, and grow (*Luo, Wang & Zeng, 2020*), and it is influenced by both environmental and physiological factors. Key photosynthetic parameters, such as net photosynthetic rate (Pn), transpiration rate (Tr), and stomatal conductance (Gs), can reflect the growth status of plants (*Gao, 2023*) and provide a foundation for deeper understanding of plants' adaptability. Stomata are essential for leaf function, acting as valves that regulate the exchange of carbon and water between plant leaves and the atmosphere. Their evolution has enabled early land plants to successfully transition from marine to terrestrial environments (*Mcadam et al., 2021*; *Wang, 2015*). The degree to which stomata open and close directly impacts vital physiological processes in plants, including transpiration, photosynthesis, and respiration (*Liu et al., 2018*). The position of stomata changes throughout the leaf growth process, gradually becoming fixed until the leaf matures (*Yang, Chen & Xian, 2023*). Stomatal characteristics not only influence the microenvironment but also respond to climate change (*Simonin & Roddy, 2018*). Stomatal traits, including density, shape, and size, result from long-term adaptations of plants to external environmental factors during evolution and are highly sensitive to environmental changes (*Zhu, Kang & Liu, 2011*). Stomatal parameters are predominantly assessed to reflect plants' responses to climate change (*Hu, Ji & An, 2015*). The leaves of angiosperms serve as the most direct and sensitive organs for sensing environmental conditions such as light, temperature, and water. The

phenotypic characteristics of leaves directly reflect the quality and growth status of the plant (*Zhang et al., 2023*). The anatomical structure of plant leaves reflects long-term adaptation to environmental conditions. Previous studies have demonstrated that variations in epidermal, palisade, and spongy tissue thickness are closely linked to photosynthetic performance and environmental responsiveness. For example, *Bacelar et al. (2004)* found that increased palisade thickness enhances photosynthetic efficiency under drought, while *Niinemets & Tenhunen (1997)* emphasized the role of leaf anatomical traits in mediating light utilization. These findings underscore the importance of leaf structural adaptations in plant ecological strategies.

*Cinnamomum camphora* L. Presl. is a significant tree species in the subtropical regions of China, primarily distributed in the southern areas of the Yangtze River, where it often occurs as an associated species in natural settings (*Leng, Wan & Liu, 2023*). The roots, stems, and leaves of *C. camphora* emit a distinct camphor aroma and exhibit various pharmacological activities including antibacterial, antioxidant, anti-inflammatory, insecticidal, analgesic, and anticancer (*Zhang et al., 2019*). *C. camphora* holds considerable potential for development and application in horticultural therapy. Additionally, *C. camphora* plays a vital role in soil and water conservation and environmental protection, which can absorb atmospheric smoke and dust and exhibit strong resistance to harmful gases such as sulfur dioxide (*Wang et al., 2024*). Current research on *C. camphora* primarily focuses on physiological changes under stress (*Zhang, Zhang & Gan, 2014*; *Zhao & Li, 2016*; *Wang et al., 2019*; *Chen et al., 2024*; *Luo et al., 2020*; *Luo et al., 2019*) and gene expression (*Liu et al., 2020*). However, studies employing common garden experiments to investigate the adaptation mechanisms of *C. camphora* from different provenances are scarce.

This study aimed to test the hypothesis that *C. camphora* seedlings from different provenances exhibit divergent physiological and anatomical traits when grown under identical environmental conditions, reflecting their provenance-specific ecological plasticity. Specifically, we asked: (1) How do photosynthetic capacity and related functional traits vary among provenances in a common garden setting? (2) Which anatomical traits contribute most to differences in photosynthetic performance? To address these questions, we conducted a common garden experiment comparing photosynthetic parameters, stomatal morphology, and leaf anatomical structures of three-year-old *C. camphora* seedlings from four different provenances. This study will provide valuable insights for the introduction of *C. camphora* in northern Chinese regions.

## MATERIALS AND METHODS

### Experimental site

The experimental site is located at Xindong Farm in the Hongqi District of Xinxiang City, Henan Province (E113°87′, N35°30′). This region has warm temperate continental monsoon climate, characterized by four distinct seasons: cold winters, hot summers, cool autumns and dry spring. Annual average temperature of approximately 14 °C. The highest recorded temperature reaches approximately 42.7 °C, whereas the lowest drops to approximately −21.3 °C. The annual average air relative humidity is 68%, and the
**Table 1** Information table of various source materials.

| Code | Seed collection site | Geographic coordinates | MAT/°C | MAP/mm | Soil type (PH) | Altitude/m |
|------|---------------------|------------------------|--------|--------|----------------|------------|
| FZ | Fujian Agriculture and Forestry University | E119°25′, N26°08′ | 12.4 | 634 | 6.5 | 6.8 |
| SH | Shanghai Botanical Garden | E121°26′, N31°10′ | 15.5 | 1,143.1 | 7.6 | 7 |
| SZ | Shenzhen Fairy Lake Botanical Garden | E114°10′, N22°34′ | 22.5 | 1,967 | 6.8 | 85 |
| WH | Huazhong Agricultural University | E114°33′, N30°47′ | 16.8 | 1,170 | 7.1 | 25.3 |

maximum depth of frozen soil is 280 mm. Annual average precipitation is 656.3 mm, with the peak occurring from June to September accounting for approximately 409.7 mm or 72% of the total annual precipitation often accompanied by heavy rainfall. The average annual sunshine duration is approximately 1,928.5 h. The frost-free period lasts approximately 220 days. The soil pH is approximately 8.2, and the surrounding vegetation primarily consists of artificially planted experimental tree species and crops.

## Experimental material

Table 1 summarizes the geographical and environmental information of the four seed provenances. These provenances–FZ (Fuzhou), SH (Shanghai), SZ (Shenzhen), and WH (Wuhan)—were deliberately selected to represent a broad range of climatic zones across subtropical China, encompassing gradients in temperature, precipitation, and soil properties. A total of 80 seedlings were examined, comprising 20 individuals from each provenance. Seeds from various provenances were germinated using a stratification treatment before cultivating in pots. On April 7, 2023, the 3-year-old seedlings were transferred to Xindong Farm under a 1 m × 1 m spacing for field maintenance. The site was pre-treated with urea and compound fertilizer. Post-transplantation management included root irrigation, monthly fertilization, mulching, weeding, and standard field care based on local weather conditions.

## Experimental design
### Leaf photosynthetic parameters

From October 28, 2023, for three consecutive days of favorable weather, the Li-6400XT portable photosynthesis measurement system (Li-Cor, Lincoln, NE, USA) was used to measure various photosynthetic parameters. Measurements were conducted hourly from 9:00 to 17:00 each day, with each measurement completed within 30 min.

For each provenance, five healthy and growth–consistent seedlings were selected. On each seedling, one fully expanded, mature, and sun-exposed leaf located in the upper third of the canopy was tagged and used consistently throughout the entire measurement period. The selected leaves were free of mechanical damage, pathogen symptoms, and shading from other organs. This approach ensured consistency in developmental stage, light exposure, and physiological condition across all measurements.

The gas parameters assessed included net photosynthetic rate (Pn), stomatal conductance (Gs), transpiration rate (Tr), and intercellular $CO_2$ concentration (Ci). Additionally, water utilization efficiency (WUE = Pn $Tr^{-1}$) and the limit value for stomatal conductance (Ls

$= 1 - Ci\ Ca^{-1}$) were calculated. The total daily carbon assimilation (mol m$^{-2}$ d$^{-1}$) was estimated by numerical integration of net photosynthetic rate (Pn) over time using the trapezoidal method, based on hourly measurements from 9:00 to 17:00.

$$A_{daily} = \sum_{i=1}^{n-1} \frac{Pn_i + Pn_{i+1}}{2} \cdot \Delta t (\Delta t = 3600s).$$

Pn and Pn$_{i+1}$: photosynthetic rate at adjacent time points
$\Delta t$: Interval between adjacent time points (in seconds)
The final result is expressed in units of $\mu$mol m$^{-2}$ d$^{-1}$. For convenience of statistics, the units are converted to mol m$^{-2}$ d$^{-1}$.

The primary real-time external factors monitored included leaf vapor pressure difference (VPD), ambient temperature (Ta), and relative humidity (RH). The instrumental environmental parameter settings were as follows: light intensity within the leaf chamber was set to 1,000 $\mu$mol m$^{-2}$ s$^{-1}$ and flow rate at 500 $\mu$mol s$^{-1}$.

Relative chlorophyll content was assessed using a SPAD-502 chlorophyll meter (Konica Minolta, Tokyo, Japan). The selected leaves were measured five times at each time point every day, and the average of all data measured over three days was used as the final analysis data.

A two-way factorial design was employed, with provenance, and hour as fixed factors in the model to account for intra—daily variation in gas exchange parameters. This design enabled a robust assessment of both provenance differences and temporal dynamics in photosynthetic responses under common garden conditions.

### Leaf stomatal characteristics

Stomatal trait measurements were conducted independently of gas-exchange measurements to avoid any interference caused by the blotting technique. The timing for stomatal measurements and the criteria for seedling selection were aligned with those for photosynthetic parameter measurements. Measurements were conducted bi-hourly from 9:00 to 17:00 daily, with each measurement completed within 30 min, using the methods by *Kübarsepp et al. (2020)* involving the blotting technique. This technique involved the use of transparent, non-elastic nail polish to capture imprints of the lower epidermis of the leaves. These imprints were mounted on a microscope slide using a tape, and observations were made using an optical microscope at 40× magnification. The epidermal imprint was observed from below, and 10 fields of view were randomly selected for image capture. ImageJ software was used to measure stomatal length (SL), width (SW), area (SS), and density (DS). For each provenance, the mean value of all measurements obtained at each time point over the 3–day period was used for final analysis.

### Anatomical structure of stems and leaves

On October 28, 2023, anatomical samples were collected from a separate group of seedlings that were not used for gas-exchange measurements, thereby ensuring that structural sampling did not interfere with the physiological observations. A total of 5–6 branch and leaf samples were taken per provenance. For leaf sampling, a segment measuring 1.0 mm

on each side of the midrib was obtained, resulting in a total length of approximately 3.0–5.0 mm. For branch sampling, a smooth, knotless section of similar length was selected. The samples were fixed (using 50% FAA fixative), dehydrated, embedded in paraffin, sliced, dewaxed, stained, and sealed to obtain complete tissue sections suitable for microscopic observation. ImageJ software was used to measure upper and lower epidermis thickness (UET, LET), palisade tissue thickness (PT), spongy tissue thickness (ST), mesophyll thickness (MT), and leaf thickness (LT). Additionally, the stem epidermal cell thickness (EpT), cortical cell thickness (CoT), xylem thickness (XyT), phloem thickness (PhT), and pith diameter (PiD) were recorded. The palisade to spongy tissue ratio (P/S (= PT/ST), cell structure compactness (CTR (= TP/LT), and cell structure looseness (SR (= ST/LT) were calculated.

### Data processing

ImageJ software was used to quantify stem and leaf anatomical structures and stomatal characteristics. Data collation and preliminary visualization were conducted using Microsoft Excel 2020. Statistical analyses were performed using SPSS 2023 (IBM Corp., Armonk, NY, USA) including one-way ANOVA followed by Tukey's HSD test ($p < 0.05$) to compare means among provenances. Pearson correlation analysis was applied to assess the relationships among photosynthetic parameters and anatomical traits. Correlation coefficients were calculated based on standardized data and visualized using Origin 2024 to generate heatmaps.

To evaluate the comprehensive adaptability of different provenances, membership function analysis was conducted using formulas (1) and (2). Xj represents the measured value, Xmin and Xmax represent the minimum and maximum values across provenances, respectively. If a positive correlation exists between the measured index and growth, membership function (1) is used; conversely, if a negative correlation is observed, the inverse membership function (2) is applied. The detailed methodology is given by *Xu et al. (2022)*.

The membership function value formula is given by:

$$U(X_j) = (X_j - X_{min})/(X_{max} - X_{min}). \tag{1}$$

The inverse membership function value formula is expressed as:

$$U(X_j) = 1 - (X_j - X_{min})/(X_{max} - X_{min}). \tag{2}$$

## RESULTS

### Environmental parameters

The diurnal variations of environmental factors at the measurement site are illustrated in Fig. 1. Both the overall air temperature (Ta) and leaf vapor pressure difference (VPD) initially exhibited an increasing trend, followed by a decrease. In contrast, relative humidity (RH) exhibited an inverse trend, decreasing as Ta and VPD increased. This inverse relationship reflects the physiological response of leaves to midday heat stress. Elevated VPD at noon corresponds with increased leaf transpiration, as water vapor loss accelerates under high temperature and low humidity conditions. Thus, VPD serves as

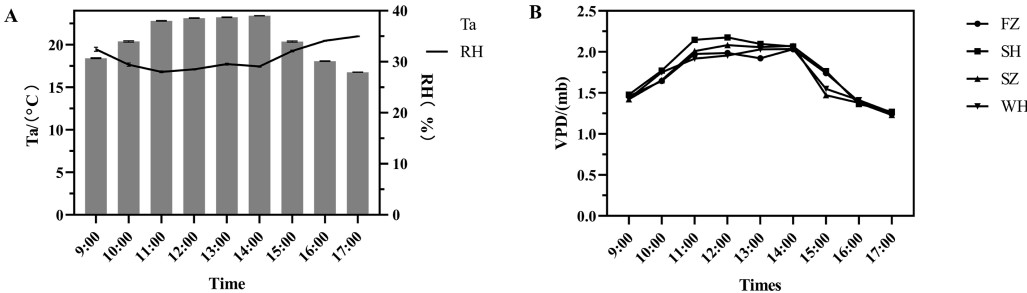

**Figure 1** **Diurnal variations of air temperature (Ta), relative humidity (RH) and vapor pressure deficit (VPD) of environmental parameters.** (A) Diurnal variations of air temperature (Ta), relative humidity (RH) (B) Diurnal variations of vapor pressure deficit (VPD) Error bars indicate mean ± SE.

a reliable indicator of the leaf's water loss and evaporative demand. Descriptive statistics (mean ± SD) were calculated for each time point, and the observed trends were used to contextualize the variation in plant physiological responses throughout the day.

## Photosynthetic characteristics

The net photosynthetic rate (Pn) of *Cinnamomum camphora* seedlings exhibited a characteristic bimodal diurnal pattern across all four provenances (Fig. 2A). While descriptive trends suggest a midday depression of photosynthesis—commonly referred to as the "photosynthetic lunch break"—particularly evident at 12:00 for FZ, SZ, and WH and at 14:00 for SH, no statistical tests were conducted to confirm the significance of these hourly fluctuations. The daily average Pn for the seedlings across four provenances followed this order: WH > FZ > SZ > SH (Table 2).

To better characterize the diurnal photosynthetic performance of *C. camphora* seedlings from different provenances, we calculated the total daily carbon assimilation per unit leaf area ($A_{daily}$) using trapezoidal integration of Pn data from 9:00 to 17:00. The results revealed that seedlings from Wuhan exhibited the highest assimilation rate (0.205 mol m$^{-2}$ day$^{-1}$), followed by Fuzhou (0.198 mol m$^{-2}$ day$^{-1}$) and Shenzhen (0.197 mol m$^{-2}$ day$^{-1}$), while those from Shanghai had the lowest (0.127 mol m$^{-2}$ day$^{-1}$) (Table 2). These findings provide a more integrative and statistically robust indicator of daily photosynthetic capacity than simple mean values or momentary peaks.

The trends in stomatal conductance (Gs) and transpiration rate (Tr) were generally similar; the seedlings from FZ, SZ, and WH exhibited low values at 12:00 and 16:00. Conversely, for the seedlings from SH, Gs and Tr peaked at 12:00 before dropping to low levels at 16:00 (Figs. 2B and 2D). The daily average Gs and Tr for the seedlings exhibited a trend of FZ > WH > SZ > SH (Table 2), with considerable differences observed between the seedlings from SH and those from FZ and WH. The intercellular carbon dioxide ($CO_2$) concentration (Ci) of all seedlings displayed a trend opposite to that of the stomatal limitation value (Ls) and water utilization efficiency (WUE) (Figs. 2C, 2E, and 2F). Specifically, Ci for the seedlings from SH peaked at 12:00, coinciding with low values of Ls and WUE. In contrast, seedlings from FZ, WH, and SZ exhibited minimum Ci value at 14:00, with Ls peaking at this time and WUE peaking at 15:00. No significant differences

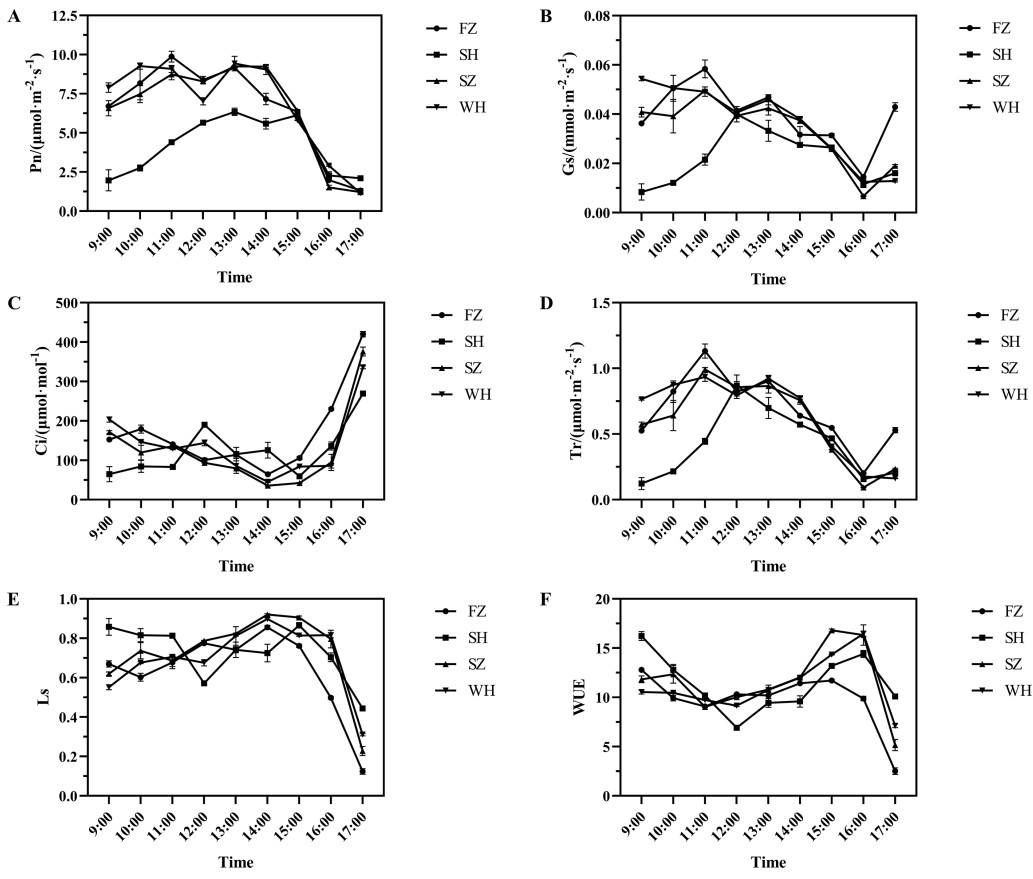

**Figure 2  Diurnal variations of stomatal gas exchange parameters in leaves of camphor seedings of different provenances.** (A) Daily variation of net photosynthetic rate; (B) Daily variation of stomatal conductance; (C) Daily variation of intercellular $CO_2$ concentration; (D) Daily variation of transpiration rate; (E) Daily variation of stomatal limitation value; (F) Daily variation of water-use efficiency. Error bars indicate mean ± SE.

**Table 2  Daily average values of gas exchange parameters in leaves of camphor seedings of different provenances.**

| Code | $P_n$ (µmol m$^{-2}$ s$^{-1}$) | $G_s$ (mmol m$^{-2}$ s$^{-1}$) | $C_i$ (µmol mol$^{-1}$) | $T_r$ (µmol m$^{-2}$ s$^{-1}$) | $L_s$ | WUE (mmol mol$^{-1}$) | SPAD | $A_{daily}$ (mol m$^{-2}$ d$^{-1}$) |
|------|------|------|------|------|------|------|------|------|
| FZ | 6.567 ± 1.003[a] | 0.039 ± 0.004[a] | 164.526 ± 35.468[a] | 0.680 ± 0.090a | 0.634 ± 0.073[a] | 9.757 ± 0.982[a] | 25.017 ± 0.510[a] | 0.198 ± 0.002[b] |
| SH | 4.134 ± 0.619[a] | 0.022 ± 0.004[b] | 123.197 ± 23.14[a] | 0.417 ± 0.087a | 0.726 ± 0.047[a] | 11.425 ± 0.972[a] | 18.650 ± 0.527[b] | 0.127 ± 0.001[c] |
| SZ | 6.522 ± 1.034[a] | 0.033 ± 0.005[ab] | 133.319 ± 34.49[a] | 0.599 ± 0.103a | 0.723 ± 0.070[a] | 11.593 ± 1.187[a] | 25.333 ± 1.920[a] | 0.197 ± 0.001[b] |
| WH | 6.843 ± 1.004[a] | 0.037 ± 0.005[a] | 146.413 ± 29.86[a] | 0.645 ± 0.104a | 0.695 ± 0.059[a] | 11.172 ± 0.930[a] | 25.667 ± 2.881[a] | 0.205 ± 0.001[a] |

**Notes.**
Different lowercase letters in the same column indicate significant differences in parameters at the 0.05 level. The same below.

were observed among the daily average Ci, Ls, and WUE of *C. camphora* seedlings across the four provenances. Additionally, the SPAD values for the seedlings from SH were significantly lower than those for the seedlings from other provenances, exhibiting a trend of WH > SZ > FZ > SH (Table 2).

**Table 3** Correlation coefficient of observed environment and camphor index.

| Index | Pn | $A_{daily}$ | Gs | Ci | Tr | Ls | WUE | SPAD |
|---|---|---|---|---|---|---|---|---|
| Ta | 0.29 | 0.29 | −0.02 | −0.44 | 0.01 | 0.67 | 0.79 | 0.31 |
| RH | 0.03 | 0.03 | 0.34 | 0.72 | 0.31 | −0.88[*] | −0.95[*] | 0 |
| VPD | −1.00[**] | −1.00[**] | −0.94[*] | −0.67 | −0.95[*] | 0.46 | 0.3 | −1.00[**] |

**Notes.**
[*] $P < 0.05$.
[**] $P < 0.01$.

The RH was significantly negatively correlated with the WUE (Table 3). Furthermore, the VPD was extremely significantly negative correlated with the Pn, $A_{daily}$ and Soil Plant Analysis Development (SPAD) value ($P < 0.01$), while exhibiting a significant negative correlation with Gs and Tr. This suggested that the WUE of camphor is a crucial factor influencing environmental humidity. Furthermore, VPD was identified as a significant environmental variable that induces diurnal variations in Pn, Gs, and Tr.

## Stomatal characteristics

The observation of the lower epidermis of *C. camphora* seedlings revealed that the stomata and guard cells exhibited a spindle shape, whereas the lower epidermal cells (LE) were square, rectangular, or round (Fig. 3). No significant difference was observed in stomatal length (SL) and stomatal width (SW) among the seedlings from different provenances (Table 4); however, the difference in stomatal area (SS) between the seedlings from SZ and FZ was more considerable. Notably, SW and SS were positively correlated. Considerable variation in stomatal density (DS) existed among the seedlings from four provenances. Specifically, no significant difference was observed in DS among the seedlings from FZ, SZ, and WH. In contrast, DS of the seedlings from SH was significantly different from that of seedlings from WH and SZ but was not significantly different from that of seedlings from FZ. DS exhibited the following trend: SZ > WH > FZ > SH, whereas SS exhibited the following trend: FZ > WH > SH > SZ. DS and SS exhibited an inverse relationship in the seedlings from FZ, WH, and SZ. A significant positive correlation was observed between SL of the seedlings and the longitude of the provenance ($P < 0.05$; Table 5). Additionally, SS was significantly correlated with annual average temperature, annual rainfall, and the altitude of the provenance. DS and provenance latitude exhibited a highly significant negative correlation ($P < 0.01$). The geographical gradient indicated that SL increases from south to north, whereas DS decreases from west to east. Furthermore, higher average temperatures, annual rainfall, and provenance altitudes were associated with smaller SS.

## Anatomical structure of the stems and leaves

The primary functions of the stem are transportation and support. In the stems, the epidermis has a protective role, whereas the cortex comprises multiple layers of parenchyma cells that contain chloroplasts for photosynthesis. The xylem is responsible for transporting water and inorganic salts, whereas the phloem transports photosynthetic products to various plant organs, with nutrients stored in the pith. In the anatomical structure of *C. camphora* stems, various tissue components exhibited notable morphological differences,

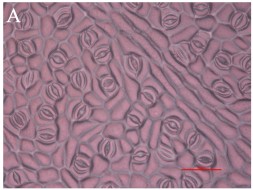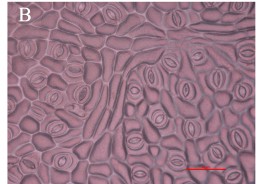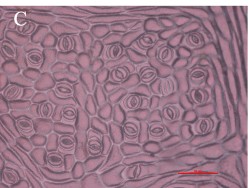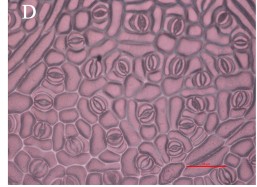

**Figure 3** **Leaf stomata morphology of camphor seedlings of different provenances.** A, B, C and D show the leaf stomata of camphor seedlings of FZ, SH, SZ and WH, respectively.

**Table 4** **Daily average value of stomatal characteristics in leaves of camphor seedlings of different provenances.**

| Code | SL (μm) | SW (μm) | SS (μm²) | DS (Per/mm²) |
|---|---|---|---|---|
| FZ | 14.091 ± 0.200[ab] | 12.789 ± 0.301[a] | 148.569 ± 5.910[a] | 719.481 ± 42.999[ab] |
| SH | 14.291 ± 0.202[ab] | 12.217 ± 0.272[a] | 141.169 ± 5.579[ab] | 654.286 ± 35.109[b] |
| SZ | 13.601 ± 0.239[b] | 11.861 ± 0.258[a] | 127.559 ± 5.169[b] | 809.091 ± 43.853[a] |
| WH | 14.635 ± 0.313[a] | 12.785 ± 0.417[a] | 146.825 ± 9.599[ab] | 784.286 ± 26.121[a] |

Notes.

Different lowercase letters in the same column indicate significant differences in parameters at the 0.05 level.

**Table 5** **Correlation coefficient of camphor stomatal characteristics and environmental factors.**

| Index | Longitude | Latitude | MAT | MAP | Altitude |
|---|---|---|---|---|---|
| SL | 0.91[*] | 0.17 | −0.53 | −0.51 | −0.72 |
| SW | 0.45 | 0.13 | −0.78 | −0.84 | −0.72 |
| SS | 0.62 | 0.4 | −0.90[*] | −0.92[*] | −0.90[*] |
| DS | −0.58 | −0.98[**] | 0.67 | 0.57 | 0.78 |

Notes.

[*] $P < 0.05$.

[**] $P < 0.01$.

and the epidermis, cortex, and stele were distinctly identifiable (Fig. 4). The lacunar and parenchyma tissues in the cortex of seedlings from different provenances were well-differentiated and developed, with vascular bundles forming distinct annular strips. The cortical cell (Co) consisted of several layers of parenchyma cells that exhibited uneven thickness among the seedlings from various sources, and the boundaries of the vascular columns were clearly defined. The thickness of the stem cortical cell (CoT), thickness of the phloem (PhT), and pith diameter (PiD) were significantly greater in the seedlings from FZ, SZ, and WH than in those from SH (Table 6). This indicated that the photosynthetic conditions of the stems of seedlings from SH were inferior to those of the stems of seedlings from other provenances, resulting in lower nutrient storage.

Plant leaves serve as the primary organs for photosynthesis. Epidermal cells of the leaves of *C. camphora* seedlings were closely arranged, featuring a visible cuticle membrane (Fig. 4). The leaf palisade tissue was situated in the upper part of the leaf (ventral surface), whereas the spongy tissue was located in the lower part (dorsal surface), characteristic of typical heterofacial leaves. The spongy tissue thickness (ST) of the leaves of *C. camphora* seedlings from four provenances was consistently thicker than the palisade tissue thickness

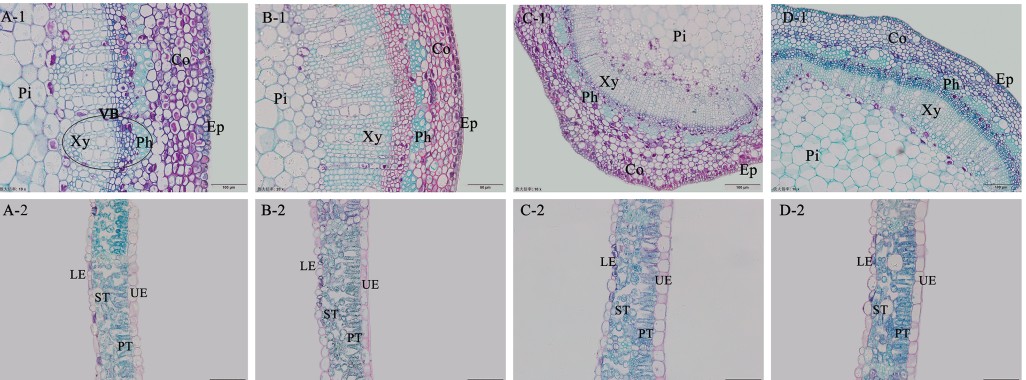

**Figure 4 Anatomical characteristics of the leaves and stems of camphor seedlings of different provenances.** A, B, C and D represent the seedlings of FZ, SH, SZ and WH, respectively. 1 and 2 represent the stem and leaf anatomical structures of camphor seedlings, respectively. Ep, epidermis cell; Co, cortex cell; Ph, phloem; Xy, xylem; VB, vascular bundle; Pi, pith; UE, upper epidermal cell; PT, palisade tissue; ST, spongy tissue; LE, lower epidermal cell.

**Table 6 Stem anatomical structural parameters of camphor seedlings of different provenances.**

| Code | EpT (μm) | CoT (μm) | PhT (μm) | XyT (μm) | PiD (μm) |
|------|----------|----------|----------|----------|----------|
| FZ | 8.12 ± 0.46[a] | 72.24 ± 3.87[a] | 66.34 ± 1.45[a] | 84.36 ± 5.42[b] | 617.23 ± 32.05[b] |
| SH | 6.91 ± 0.28[a] | 51.45 ± 2.12[c] | 38.27 ± 1.94[c] | 114.80 ± 3.63[a] | 538.61 ± 9.75[c] |
| SZ | 8.02 ± 0.35[a] | 58.92 ± 1.39[b] | 45.70 ± 2.83[b] | 79.30 ± 2.78[b] | 592.09 ± 15.57[bc] |
| WH | 8.16 ± 0.53[a] | 60.47 ± 2.12[b] | 49.35 ± 2.23[b] | 88.79 ± 5.28[b] | 748.83 ± 31.81[a] |

**Notes.**

Different lowercase letters in the same column indicate significant differences in parameters at the 0.05 level.

(PT) (Table 7). The palisade tissue, palisade to spongy tissue ratio (P/S), and structural compactness (CTR) of *C. camphora* seedlings from WH and SZ were significantly thicker than other provenances, indicating that their leaf structural characteristics were more conducive to photosynthesis. In contrast, the upper epidermal thickness (UET), PT, mesophyll cell thickness (MT), leaf thickness (LT), P/S, and CTR of the seedlings from SH were thinner than those of the seedlings from other provenances, suggesting that the leaf photosynthetic structures of the seedlings from SH were somewhat less effective.

The latitude of the provenance was significantly negatively correlated with UET ($P < 0.05$) and was extremely significantly negatively correlated with the P/S and CTR ($P < 0.01$; Table 8). Additionally, it was very significantly positively correlated with structural looseness (SR). From west to east, the geographical gradient exhibited a gradually decreasing trend for UET, P/S, and CTR and a gradually increasing trend for SR. Furthermore, annual rainfall and ST were significantly negatively correlated ($P < 0.05$), and soil pH and LET were significantly positively correlated ($P < 0.05$).

## Correlation analysis

Using Pearson correlation analysis, we explored the functional relationships among photosynthetic, anatomical, and structural traits in *C. camphora* seedlings (Fig. 5).

**Table 7  Leaf anatomical structural parameters of camphor seedlings of different provenances.**

| Code | UET (µm) | PT (µm) | ST (µm) | LET (µm) | MT (µm) |
|---|---|---|---|---|---|
| FZ | 14.92 ± 0.61[b] | 23.22 ± 0.49[b] | 42.51 ± 1.12[a] | 11.07 ± 0.24[b] | 65.86 ± 2.44[a] |
| SH | 13.06 ± 0.52[c] | 18.28 ± 0.69[c] | 36.18 ± 1.24[b] | 13.18 ± 0.63[a] | 54.93 ± 1.10[b] |
| SZ | 15.67 ± 0.35[b] | 24.24 ± 0.89[ab] | 34.93 ± 1.18[b] | 11.12 ± 0.68[b] | 58.03 ± 1.07[b] |
| WH | 17.34 ± 0.75[a] | 26.14 ± 0.96[a] | 38.22 ± 1.32[b] | 11.44 ± 0.45[b] | 62.68 ± 0.66[a] |

| Code | LT (µm) | P/S (%) | CTR (%) | SR (%) |
|---|---|---|---|---|
| FZ | 85.98 ± 2.00[a] | 0.55 ± 0.017[b] | 0.27 ± 0.007[bc] | 0.50 ± 0.007[a] |
| SH | 75.41 ± 0.63[c] | 0.51 ± 0.032[b] | 0.25 ± 0.009[c] | 0.49 ± 0.021[a] |
| SZ | 79.39 ± 1.09[b] | 0.70 ± 0.033[a] | 0.30 ± 0.011[a] | 0.43 ± 0.015[b] |
| WH | 88.13 ± 0.68[a] | 0.67 ± 0.030[a] | 0.29 ± 0.010[ab] | 0.44 ± 0.018[b] |

Notes.
Different lowercase letters in the same column indicate significant differences in parameters at the 0.05 level.

**Table 8  Correlation coefficient of environmental conditions in provenances and leaf anatomical characters of camphor seedlings.**

| Index | UET | PT | ST | LET | MT | LT | P/S | CTR | SR |
|---|---|---|---|---|---|---|---|---|---|
| Longitude | −0.15 | −0.35 | 0.05 | 0.69 | −0.17 | 0.01 | −0.47 | −0.62 | 0.29 |
| Latitude | −0.88* | −0.87 | 0.3 | 0.58 | −0.19 | −0.44 | −1.00** | −0.96** | 0.97** |
| MAT | 0.3 | 0.29 | −0.84 | −0.07 | −0.5 | −0.32 | 0.77 | 0.66 | −0.8 |
| MAP | 0.2 | 0.16 | −0.90* | 0.06 | −0.61 | −0.43 | 0.69 | 0.55 | −0.74 |
| PH | −0.41 | −0.61 | −0.58 | 0.96* | −0.78 | −0.58 | −0.32 | −0.57 | 0.11 |
| Altitude | 0.36 | 0.42 | −0.63 | −0.36 | −0.26 | −0.18 | 0.81 | 0.79 | −0.77 |

Notes.
*$P < 0.05$.
**$P < 0.01$.

In addition to instantaneous net photosynthetic rate (Pn), we also included the integrated metric of daily carbon assimilation per unit area ($A_{daily}$) to better capture diurnal photosynthetic performance. Pn and $A_{daily}$ exhibited highly similar correlation patterns, both showing strong positive associations with SPAD ($r = 1.00$), Gs ($r = 0.95$), Tr ($r = 0.96$), and PT ($r = 0.96$), underscoring that chlorophyll content, stomatal conductance, transpiration, and palisade tissue thickness are critical for photosynthetic capacity. The perfect correlation ($r = 1.00$, **$p < 0.001$) between SPAD, Pn and $A_{daily}$ highlights the validity of SPAD as a proxy for chlorophyll content and cumulative photosynthetic performance. Gs and Tr were tightly correlated ($r = 1$, **$p < 0.001$), reflecting the physiological linkage between stomatal aperture and gas exchange. Gs also negatively correlated with LET ($r = −0.95$, $p < 0.05$), suggesting that reduced lower epidermis thickness may facilitate stomatal gas exchange, consistent with anatomical observations. Ci was positively associated with ST ($r = 0.92$) and negatively with WUE ($r = −0.90$), indicating that a looser spongy mesophyll favors $CO_2$ diffusion at the cost of water-use efficiency. Similarly, Tr was positively correlated with SPAD ($r = 0.95$) and negatively with WUE ($r = −0.58$), reinforcing the trade-off between carbon gain and water loss under varying physiological conditions. Among stem traits, CoT and PhT were strongly correlated ($r = 0.99$, **$p < 0.01$), and PiT was significantly associated with UET ($r = 0.92$),

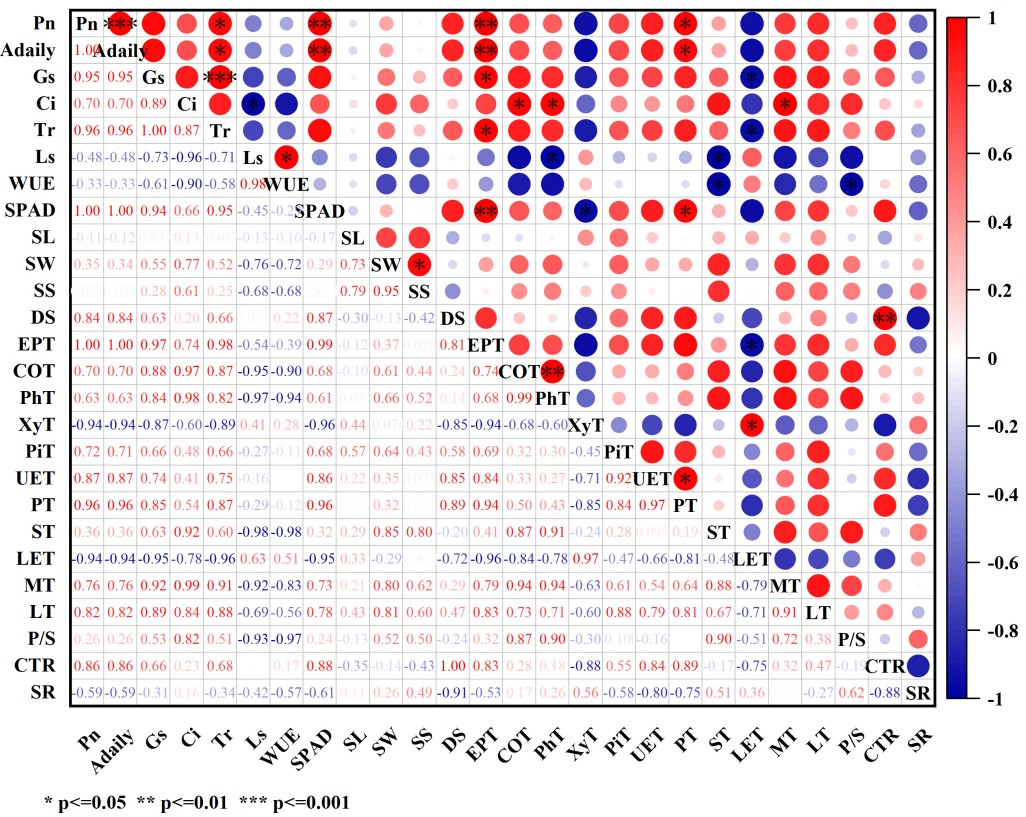

* p<=0.05   ** p<=0.01   *** p<=0.001

**Figure 5** Heat map of the correlation between photosynthetic characteristics and anatomical structure. *$p <= 0.05$, **$p <= 0.01$, ***$p <= 0.001$.

suggesting a functional coupling between stem transport capacity and upper epidermis development. Together, these correlations demonstrate that traits related to leaf structure (PT, ST, LET), stomatal function (Gs), and pigment content (SPAD) jointly regulate both instantaneous and daily carbon assimilation in *C. camphora* seedlings, shaping their adaptive photosynthetic strategies.

## Comprehensive evaluation of the photosynthetic capacity

To thoroughly assess the advantages and disadvantages of *C. camphora* seedlings from different provenances, we applied membership function analysis based on a set of physiological and anatomical indicators. Prior to this analysis, Pearson correlation analysis was conducted to determine the relationship between each indicator and photosynthetic capacity (Pn). Indicators that showed significant negative correlations with Pn—such as stomatal limitation value (Ls), water-use efficiency (WUE), xylem thickness (XyT), lower epidermis thickness (LET), and cell structure looseness (SR)—were considered as inverse indicators and calculated using the inverse membership function formula. In contrast, positively correlated indicators were assessed using the standard membership function formula. The average membership function values of *C. camphora* seedlings from different provenances followed the order WH (0.692) > FZ (0.558) > SZ (0.457) > SH (0.378) (Table 9). While WH and FZ exhibited relatively higher overall performance scores, the

**Table 9 Comprehensive evaluation table of the index membership function of different provenances.**

| Index | FZ | SH | SZ | WH |
|---|---|---|---|---|
| Pn | 0.898 | 0.000 | 0.881 | 1.000 |
| $A_{daily}$ | 0.910 | 0.000 | 0.897 | 1.000 |
| Gs | 1.000 | 0.000 | 0.660 | 0.848 |
| Ci | 1.000 | 0.000 | 0.245 | 0.562 |
| Tr | 0.000 | 1.000 | 0.307 | 0.133 |
| Ls | 1.000 | 0.000 | 0.039 | 0.334 |
| WUE | 0.000 | 0.908 | 1.000 | 0.771 |
| SPAD | 0.093 | 1.000 | 0.048 | 0.000 |
| SL | 0.474 | 0.667 | 0.000 | 1.000 |
| SW | 1.000 | 0.380 | 0.000 | 0.986 |
| SS | 1.000 | 0.648 | 0.000 | 0.917 |
| DS | 0.421 | 0.000 | 1.000 | 0.840 |
| EpT | 0.973 | 0.000 | 0.894 | 1.000 |
| CoT | 1.000 | 0.000 | 0.360 | 0.434 |
| PhT | 0.000 | 1.000 | 0.735 | 0.605 |
| XyT | 0.142 | 1.000 | 0.000 | 0.267 |
| PiD | 0.374 | 0.000 | 0.254 | 1.000 |
| UET | 0.435 | 0.000 | 0.609 | 1.000 |
| PT | 0.629 | 0.000 | 0.758 | 1.000 |
| ST | 0.000 | 0.836 | 1.000 | 0.567 |
| LET | 0.000 | 1.000 | 0.140 | 0.274 |
| MT | 1.000 | 0.000 | 0.284 | 0.709 |
| LT | 0.831 | 0.000 | 0.313 | 1.000 |
| P/S | 0.216 | 0.000 | 1.000 | 0.863 |
| CTR | 0.542 | 1.000 | 0.000 | 0.197 |
| SR | 0.000 | 0.029 | 1.000 | 0.972 |
| Average membership function value | 0.558 | 0.378 | 0.457 | 0.692 |
| Rank | 2 | 4 | 3 | 1 |

differences between them may not be statistically significant and should be interpreted as indicative rather than conclusive. In contrast, SH consistently showed the lowest average membership function value, suggesting relatively weaker comprehensive performance.

# DISCUSSION

Stomatal closure is recognized as a primary factor contributing to decline in Pn (*Figueras et al., 2004*). In this study, Pn of *C. camphora* seedlings from four provenances generally showed a "double-peak" diurnal pattern. The temporal trends of Pn, Gs, and Tr appeared broadly similar (Figs. 2A, 2B, and 2D), and Pearson correlation analysis indicated positive relationships among the these parameters (Fig. 5). Total daily carbon assimilation per unit leaf area ($A_{daily}$) differed substantially among provenances (Table 2), reflecting variation in photosynthetic capacity and physiological adaptation. The superior performance of Wuhan and Fuzhou seedlings may be attributed to their higher Pn and Gs values, which

were supported by their favorable leaf structural traits. In contrast, the substantially lower $A_{daily}$ of Shanghai seedlings is consistent with their reduced Pn, likely resulting from stomatal limitations and potentially higher sensitivity to environmental stress. This suggested a synergistic relationship between stomatal opening and Pn. Both stomatal and non-stomatal limitations influence photosynthesis (*Rao & Chaitanya, 2016*). *Farquhar & Sharkey (1982)* suggested that a decrease in Pn, accompanied by an increase in Ls and decrease in Ci, indicates that the decline in Pn is primarily due to stomatal factors; conversely, if Ls decreases and Ci increases, non-stomatal factors become the main contributors to the reduction in Pn. In *C. camphora* seedlings from FZ, SH, and SZ, Ls increased but Ci decreased at noon, suggesting that stomatal closure was the principal reason for the decline in Pn. In contrast, Wuhan seedlings showed decreased Ls and increased Ci, suggesting that biochemical or photochemical factors—rather than stomatal closure—limited photosynthesis at noon. These findings suggest distinct physiological adjustment strategies among provenances in response to transient environmental stress. Although all seedlings were grown in a common garden to minimize environmental variation, their original source environments differ: Fuzhou and Wuhan are characterized by subtropical monsoon climates with ample rainfall and moderate summer temperatures, while Shanghai experiences frequent summer heat and humidity, which may predispose seedlings to greater midday photoinhibition. In this context, we define "adaptability" as the capacity of seedlings to maintain high photosynthetic performance and efficient carbon–water tradeoffs under transient stress conditions. The lower $A_{daily}$ and reduced midday Pn in Shanghai seedlings reflect a limited ability to sustain gas exchange during peak environmental demand, indicative of lower physiological adaptability. This reduced capacity for carbon assimilation may ultimately constrain biomass accumulation and ecological competitiveness under similar climatic conditions (*Nikinmaa et al., 2013*). The stomata serve as the primary channels for two essential processes: entry of $CO_2$ into the mesophyll cells during photosynthesis and water loss through transpiration. The efficiency of gas exchange is determined by DS and SS (*Sack & Buckley, 2016*). In this study, the short axis of the stomata and SS were significantly positively correlated (Fig. 4), suggesting that SS is primarily influenced by the angle of stomatal opening and closing. Differences in SS and DS were observed among the seedlings from four provenances (Table 3). Notably, except for SH, smaller SS correlated with greater DS in other three provenances. This is consistent with the results of *Cao et al. (2020)* regarding mulberry stomata. Specifically, SS exhibited a trend of FZ > WH > SZ, whereas DS exhibited a trend of SZ > WH > FZ. The SW and SS of *C. camphora* seedlings from FZ and WH were higher, suggesting that they were better at gas exchange than the seedlings from other provenances. WUE, defined as the dry mass produced per unit mass of water consumed, reflects the relationship between leaf water consumption and material accumulation (*Wu, Tian & Xie, 2020*). Pn of *C. camphora* seedlings from FZ was lower than that of seedlings from WH and SZ. However, the seedlings from FZ exhibited the highest Tr; consequently, their WUE was the lowest among the seedlings from four provenances, indicating greater photosynthetic water requirement and higher overall tolerance, although with reduced drought resistance. The photosynthetic capacity of plants is closely linked to the content of chlorophylls,

which not only capture and transmit light energy but also facilitate its conversion (*Hu et al., 2010*). The relative chlorophyll content of *C. camphora* seedlings exhibited a trend of WH > SZ > FZ > SH (Table 2), and the chlorophyll content was observed to be extremely significantly positively correlated with Pn.

Compared with the seedlings from SH, the CoT, PhT, and PiD of the stems of *C. camphora* seedlings from FZ, SZ, and WH were significantly higher, facilitating the accumulation of photosynthetic pigments. The accumulation of photosynthetic products is linked to transportation. Therefore, the photosynthetic conditions in FZ, SZ, and WH were superior to those in SH. Changes observed in the upper and lower epidermises of leaves and P/S provide evidence of the plants' ability to adapt to varying environmental conditions (*Oliveira et al., 2018*). In this study, the ST was higher than the PT under natural conditions without stress, which is consistent with the findings of *Afas, Marron & Ceulemans (2007)* in poplar leaves. Chloroplasts are predominantly located in palisade tissue of leaves, which are closely linked to photosynthetic efficiency (*Silva & Santos, 2023*). In our study, PT was significantly positively correlated with both Pn and SPAD values. Leaf anatomical features, including the epidermis, palisade, and spongy parenchyma, are finely tuned to optimize WUE and photosynthesis (*Ackerson & Krieg, 1977*). In our study, ST was significantly negatively correlated with Ls and WUE and significantly positively correlated with Ci. Additionally, LET was significantly negatively correlated with Pn, Tr, and SPAD values (Fig. 5). Palisade tissue serves as the primary site for chloroplasts in plant leaves and is integral to leaf photosynthesis. ST is located adjacent to lower epidermis, with the interstitial spaces within ST serving as the primary sites for $CO_2$ storage in plants. Consequently, thicker ST leads to enhanced $CO_2$ storage, which is intricately linked to photosynthesis and associated reactions. It is evident that the parameters of PT and ST are critical indicators influencing various photosynthetic metrics. In this study, the leaf PT, P/S, and CTR of seedlings from WH and SZ were significantly higher than those of the seedlings from other provenances, suggesting that the seedlings from WH and SZ possess superior photosynthetic structures. Conversely, UET, PT, MT, LT, P/S, and CTR of seedlings from SH were observed to be the lowest, indicating a less effective photosynthetic structure.

Based on the anatomical examination of the stems and leaves of *C. camphora* seedlings from the four provenances, it can be concluded that the photosynthetic structure of seedlings from SH was the least favorable. Furthermore, this study reported significant or extremely significant positive correlation between CTR and both DS and P/S, whereas SR was significantly negatively correlated with DS and P/S. It was speculated that PT and ST also influence plant photosynthesis by affecting DS.

Plants' adaptation to the environment is a complex process. The structure of leaf tissue is sensitive to surrounding environmental conditions, and leaf morphological characteristics are closely related to plant growth strategies and resource utilization (*Yu et al., 2019*). In this study, significant or extremely significant differences were observed in the anatomical structure of *C. camphora* leaves (specifically, UET, P/S, CTR, and SR) across different geographical distribution areas, indicating that these plants have undergone local adaptation. However, the correlation between the anatomical structure of *C. camphora* leaves and other environmental factors of the original provenance appears

weak; this suggested that though environmental factors influence *C. camphora* leaves, their contribution is relatively minor (Table 8). Moreover, all original provenances of *C. camphora* have subtropical monsoon climate, where there has been no significant change in the climatic environment. Our findings are consistent with those of *Chen et al. (2019)* who examined changes in the leaf functions of *Machilus pauhoi* seedlings from various provenances under common garden environments. Despite the adaptability of the test materials, which have undergone 3 years of growth in a common garden, notable differences in the anatomical structure of leaves persist among various sources. This variability may arise from the varying degree of adaptability to garden environment exhibited by the seedlings.

It is important to acknowledge that while ecophysiological traits and anatomical structures provide valuable insights into the adaptability of *C. camphora*, they alone may not offer a complete picture of provenance-specific adaptation. Environmental adaptability is influenced by a combination of intrinsic factors (*e.g.*, genetic background, leaf morphology, stomatal traits) and extrinsic factors, including edaphic characteristics and management practices such as nutrient input during seedling establishment. Although this study was conducted under common garden conditions to minimize environmental variability, we recognize that the original provenance environments—including soil types, nutrient availability, and climatic legacies—may still influence phenotypic expression. Therefore, future research should integrate edaphic data and standardized nutrient application protocols to better elucidate the relationships between trait variation and intrinsic adaptability. Such multifactorial approaches will help refine provenance selection for ecological restoration and afforestation efforts.

## CONCLUSIONS

The ability of a single index to evaluate the adaptability of plants is limited. Therefore, we performed a comprehensive evaluation based on photosynthetic performance (*e.g.*, Pn, Gs, WUE, $A_{daily}$) and anatomical traits (*e.g.*, PT, ST, SPAD) of *C. camphora* seedlings. The integrative results (Table 9) indicated that seedlings from the SH provenance exhibited overall lower photosynthetic efficiency and less favorable leaf structural characteristics under the same environmental conditions, which may compromise their physiological plasticity and potential for biomass accumulation relative to other provenances. These traits suggest that SH seedlings may be less competitive in resource acquisition (*e.g.*, carbon assimilation) under high-temperature and high-light conditions, although direct measurements of growth or reproductive fitness were not conducted in this study. In contrast, the seedlings from FZ and WH exhibited strong photosynthetic capacity, superior stem and leaf structure, that enhances the plants' stress resistance. These seedlings exhibited larger SW and SS, and moderate SD which is advantageous for gas exchange. Collectively, these traits were observed to promote photosynthesis, increase organic matter accumulation, enhance overall properties, and improve competitiveness compared with the seedlings from other provenances (Table 9). Leaf PT, P/S, CTR, and chlorophyll content were the critical determinants of the photosynthetic capacity of *C. camphora* seedlings

across the four provenances. Though variations were observed in the adaptability of *C. camphora* seedlings from different provenances to the same environment, the differences in leaf structure among the seedlings from provenances with similar climatic conditions were not significantly influenced by the environmental factors of their original provenance. This suggested a potential relationship with their intrinsic adaptability. Our findings provided a vital theoretical foundation for the introduction and cultivation of *C. camphora* in various regions and provided insights for the selection of provenances and conservation of genetic resources in the future.

### Funding
The funding was supported by the Henan Province Science and Technology Plan Project (Grant No. 162102110155), and Key Scientific Research Project of Henan Province Colleges and Universities (Grant No. 15A220002), and Xinxiang City Science and Technology Innovation Development Special Project (Grant No. CXGG16029). The funders had no role in study design, data collection and analysis, decision to publish, or preparation of the manuscript.

### Grant Disclosures
The following grant information was disclosed by the authors:
The Henan Province Science and Technology Plan Project: 162102110155.
Key Scientific Research Project of Henan Province Colleges and Universities: 15A220002.
Xinxiang City Science and Technology Innovation Development Special Project: CXGG16029.

### Competing Interests
The authors declare there are no competing interests.

### Author Contributions
- Yufeng Liu conceived and designed the experiments, performed the experiments, analyzed the data, prepared figures and/or tables, authored or reviewed drafts of the article, and approved the final draft.
- Ruobing Zou performed the experiments, prepared figures and/or tables, and approved the final draft.
- Yanmeng Guo performed the experiments, authored or reviewed drafts of the article, and approved the final draft.
- Yu Ji conceived and designed the experiments, prepared figures and/or tables, and approved the final draft.
- Mengyuan Li analyzed the data, authored or reviewed drafts of the article, and approved the final draft.
- Yang You conceived and designed the experiments, analyzed the data, authored or reviewed drafts of the article, and approved the final draft.

## Data Availability

Raw data is available in the Supplemental Files.

## Supplemental Information

Supplemental information for this article can be found online at http://dx.doi.org/10.7717/peerj.19934#supplemental-information.

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
