# Peer review of "Elucidation of the adaptability of Cinnamomum camphora seedlings from different provenances via analyzing photosynthetic characteristics and anatomical structure"

_PeerJ, doi:10.7717/peerj.19934_

## Round 0.1 · original submission · Major Revisions

Dear authors, the work is not devoid of content and merit, but it needs to be reorganized appropriately. Two of the reviewers are very critical and suggest profound changes, and I share their opinion. First of all, the scientific hypothesis and the objectives must be clearly stated at the end of the introduction. The choice of the 4 provenances and why it is important to introduce this plant in gardens are not clear. Also, concepts relevant to the experiment, such as adaptability and competitiveness must be defined clearly and commented in the discussion. The statistics also need to be more detailed: the sample size, the parametric/non-parametric test and the type of pot-hoc comparison applied are not clear in the tables and figures. I invite the authors to follow all the indications of the reviewers and modify the text and figures accordingly.

**Language Note:** The review process has identified that the English language must be improved. PeerJ can provide language editing services - please contact us at [email protected] for pricing (be sure to provide your manuscript number and title). Alternatively, you should make your own arrangements to improve the language quality and provide details in your response letter. – PeerJ Staff

Reviewer 1 ·

Basic reporting

No comment

Experimental design

The experimental design is not clear at all and proper statistical analyses do not appear to have been conducted. See additional comments.

Validity of the findings

It is not clear to me what the hypothesis to be tested was. The authors assessed traits of the species from four different provenances in a fifth environment common garden experiment, but what was the scientific question that was addressed?

Again, I have problems with the experimental design and analyses.

Additional comments

Line 16: I don’t know what is meant by “to provide insights into the practice of garden introduction”.

There is no clear explanation of why the four provenances were chosen. What was the hypothesis that was tested?

How were leaves for measurements selected? How did the authors ensure that consistent leaves (age, canopy position, etc.) were chosen for measurements?

The data collection is not clear. The authors state that gas-exchange measurements were made hourly for three consecutive days on 5 seedlings. Was the same leaf used on the same seedling each day or were different leaves used? Either way, a full statistical analysis that includes accession, day, and hour should be presented. Depending on what was actually done, the models will be quite different.

Again, the stomatal trait measurements are not clear. My understanding as written is that samples were collected every two hours (line 137). How does this correspond to leaves that were used for gas-exchange measurements? If the same leaf was used throughout the day, why would multiple measurements of epidermal traits be necessary? Also, if this were the case, the technique used to quantify epidermal traits would render those leaves unusable for subsequent gas-exchange measurements.

Line 147 states that branches and leaves were removed from seedlings. This is the first day of the gas-exchange measurements and will certainly have affected those traits.

Line 188: The authors claim that the SH had lower Pn rates at noon (and the abstract suggests that all accessions exhibited this pattern), but there are no statistics to justify this claim. Related to this, the authors make claims throughout about low/high points for various measurements throughout the day, but again, these claims are not vali without statistics showing this is the case. I am guessing there is sufficient variation in their measurements that mathematically lower/higher values are not statistically significant in many cases.

If the point was to quantify diurnal patterns of Pn rates, does it really matter what the average was?
Why not calculate the total daily C assimilation per unit area instead?

Line 267: The corr heat map is not Fig 4.

Many of the correlations are not relevant. For example, of course all gas-exchange parameters are going to be correlated, as they are measured using a limited set of parameters. The authors should identify correlations that may have physiological/functional significance and focus on these. Section 2.5 is simply a laundry list of correlations and not very insightful.

Line 304: is FZ really different from WH?

Line 310-311: Again, there is no statistical basis for this claim.

Line 325-326: The authors need to provide more detail on what they mean by “adaptability”. How are the four provenances different from the common garden site?

Lines 401-403: The authors do not define “competitiveness”. Are they referring to growth (for which there is no quantification)? Fitness? The accessions have different traits, but there is no link to these traits and the ability to “compete” with other accessions.

Table 1: Information should be provided for the common garden site.

Figure 1 should be supplemental.

Error bars should be included on figures.

·

Basic reporting

The paper has potential for publication, however, it lacks important information for acceptance for publication. The introduction does not contextualize the central problem of the paper, there are no hypotheses and the objective is not clear. The methodology lacks several pieces of information about how the paper was carried out. The main problem is the statistical analysis performed; in my opinion, another analysis should be performed to compare the four sources. Since the analysis used in the paper is not consistent with what the authors wanted to demonstrate, the results and discussion are scattered throughout the manuscript.

Experimental design

What is the sample size? This information needs to be included. There is a lot of missing information in the methodology. Mainly how many individuals were measured. Include a topic about the statistical analyses. Explain in detail which analyses were performed. I could not understand which analysis was performed. I would like to emphasize the importance of detailing the statistical analysis. I suggest that you compare the four different sources. Perhaps a one-way ANOVA would be more appropriate.

Validity of the findings

no comment

Additional comments

Introduction
The introduction lacks contextualization of the paper's theme. The authors need to improve the definition of the problem. The paper has no hypotheses. Improving the contextualization of the introduction will help the authors to support the hypotheses. What is the problem of the paper? What is the central question of the paper? The objective of the paper needs to be clear.
Results
I cannot understand the results because I did not understand the statistical analysis performed. Include the captions inside the graphs (e.g. Ta, RH).
Discussion
The discussion gives a general idea of what happened to the plants in the study, however, the authors did not compare four sources (note: at least the text does not indicate this), so it should be rewritten after the suggested corrections.
Conclusion
You cannot conclude this way. Not without the other suggested corrections.

Reviewer 3 ·

Basic reporting

• The manuscript is clear and well written using simple English language.
• Most of the point are well specified to have a full understanding.
• Introduction part is good. But it could have improved by providing some more information regarding the species and its utility.
• Much of the information included is about the ecophysiology. Whereas information related to species, extent has to be incorporated.
• Literature of review is adequate and well placed. It is relevant and specific to the topic.
• The structure of the manuscript conforms the peer standard.
• Figures are relevant well labelled description could have improved.

Experimental design

• The research work is original. There were similar kind of studies carried out in different part of the world.
• Research question is well defined, relevant and meaningful.
• Author has stated the research gap and how this study could help to fill it.
• Sophisticated instruments were used for the research, investigation was good and it followed technical and ethical standards
• Methods described by sufficient details to obtain a result.
• The number of replications could have increased for better clarity

Validity of the findings

• Results were presented well with the support of figures and tables
• Discussion part was good substantiated with proper reference
• Conclusions are well stated, linked to original research question.
• There are certain points to be consider those are- ecophysiology and anatomical structure alone not sufficient to answer the adaptability to a particular environment-
• Edaphic character of the area is not discussed, it can be also a factor to be considered-
• As the author stated single index to evaluate adaptability is not possible-
• Effect of nutrient application during the planting time also can influence the ecophysiology-
• Variation obtained here can be used as a potential relationship with intrinsic adaptability

Additional comments

• Manuscript is well written with sufficiently large quantity of information-
• Criteria of selection of provenances needs to be addressed
• Incorporate edaphic characteristics of the provenance in details-
• Incorporate the details of the nutrient application during the planting operation since it can be influence the physiology-
• Heat map of correlation between photosynthetic character and anatomical structure is a plus point

---

## Round 0.2 · accepted · Accept

The Authors addressed all points raised by the Reviewers, incorporating the additional information requested in the revised version, and making the due changes in the text and figures. I myself have checked point by point, and I approve the changes made. Therefore, the manuscript can be accepted in the current version.

·

Basic reporting

The authors followed all suggested corrections.

Experimental design

The changes made by the authors made the sampling procedure used much clearer.

Validity of the findings

The article has interesting findings.

Additional comments

No comments.